# Beyond Connectivity: A Multi-Dimensional AI Readiness Gap Analysis for Northeast India

*A Conceptual Framework Paper – North East Gen AI 2026*

Track 3: AI and Technology for NE India

Baksheesh Sachar, Department of Planning & Investment, Government of Arunachal Pradesh

Amme Shirisha, Department of Planning & Investment, Government of Arunachal Pradesh

## Abstract

Northeast India, home to over 200 languages and more than 46 million people — occupies a structurally marginal position in the global AI landscape. Existing frameworks, designed for nation-states, fail to capture the sub-national, multi-ethnic, and linguistically hyperdiverse conditions of this region. This paper proposes a six-dimensional **AI Readiness Gap Framework for Northeast India (ARGF-NE)**: (1) digital connectivity infrastructure, (2) linguistic data availability, (3) compute access, (4) human capital, (5) policy alignment, and (6) institutional capacity. For each dimension, we identify critical gaps and propose targeted interventions. We argue Northeast India constitutes a uniquely important testbed for low-resource, high-diversity AI with insights that extend to similarly positioned regions globally.

**Keywords:** AI readiness, Northeast India, low-resource NLP, digital divide, linguistic diversity, AI policy

## 1 Introduction

Northeast India comprises eight states Assam, Meghalaya, Manipur, Mizoram, Nagaland, Arunachal Pradesh, Tripura, and Sikkim accounting for approximately 3.7% of India's population and one of the highest concentrations of linguistic diversity on earth. The region holds a paradoxical position in the AI era: its extraordinary richness in languages makes it a prime candidate for AI-assisted documentation, while its structural conditions — terrain, connectivity gaps, limited compute access, and sparse NLP resources place it among the least AI-ready territories in Asia.

Global AI readiness assessments, including the Oxford Government AI Readiness Index 2025 and the World Bank's Digital Progress and Trends Report 2025, evaluate countries through metrics calibrated for nation-states with centralised infrastructure and high-resource data environments. None accounts for sub-national heterogeneity like Northeast India's: eight states, over 200 languages, uneven connectivity, and policy ecosystems that frequently bypass the region's specific needs.

This paper makes three contributions: it proposes the **ARGF-NE** framework; synthesises publicly available evidence across each dimension; and offers targeted interventions for researchers, practitioners, and policymakers. The paper is exploratory, generated with AI assistance (disclosed in Section 5), and does not claim primary empirical data.

## 2 Background

The World Bank's 2025 report identifies four foundational AI pillars compute, data, talent, and coordination as prerequisites for meaningful AI participation. The Oxford AI Readiness Index 2025 assesses 193 countries; top performers share robust digital infrastructure, large datasets, and active AI strategies. India's IndiaAI Mission signals national ambition, but its translation to peripheral sub-national regions like the Northeast remains limited.

Northeast India's connectivity deficit is documented and severe. As of 2025, the region holds **under 2% of India's 974 million broadband subscribers** despite representing 3.7% of the population. BharatNet penetration is highly uneven (Assam: 1,507 gram panchayats connected; Sikkim: 35). The Siliguri Corridor only 22 km wide creates a geographic data bottleneck, exacerbated by terrain-induced disruptions from floods and landslides.

The linguistic landscape compounds this challenge. Over 200 languages are spoken in the region, of which fewer than ten have usable NLP corpora. Assamese has seen growing research attention (AxomiyaBERTa, Tamang & Bora 2024, Nath et al. 2025), but the vast majority of languages such as Mising, Rabha, Tiwa, Karbi, Dimasa, Adi, and hundreds more have **no digital presence whatsoever**, rendering them invisible to every existing LLM.

## 3  The ARGF-NE Framework

We propose the AI Readiness Gap Framework for Northeast India (ARGF-NE), a six-dimensional instrument adapted from global readiness literature (World Bank 2025; Oxford Insights 2025; ITU GAIN 2024) but reconfigured around dimensions most structurally determinative for the Northeast. Table 1 provides the consolidated overview; narrative detail for each dimension follows.

| Dimension | Key Gaps | Evidence | Priority | Intervention (summary) |
|---|---|---|---|---|
| **1. Connectivity** | <2% of national broadband; 12.73M active mobile users; Siliguri Corridor bottleneck | ORF 2025; TRAI data | Critical | Satellite last-mile; AI infrastructure zone under PM DevINE |
| **2. Linguistic data** | 200+ languages; <10 with usable NLP corpora; invisible to all LLMs | Gupta 2025; Tamang & Bora 2024 | Critical | NE Language Data Commons; AI4Bharat partnership |
| **3. Compute access** | No regional data centres; reliance on remote cloud; power irregularities | World Bank 2025 | High | Regional GPU Hub at IIT Guwahati/NIT Silchar |
| **4. Human capital** | ICT lab decline (Assam 2021–23); <33% digital literacy; sparse AI curriculum | ORF 2025 | High | NCF 2023 AI literacy; university Centres of Excellence |
| **5. Policy alignment** | Digital NE Vision 2022 lacks AI/LLM strategy; IndiaAI Mission not sub-nationally targeted | MeitY 2022; Oxford 2025 | Moderate | Northeast AI Addendum to IndiaAI Mission |
| **6. Institutional capacity** | Fragmented research; thin industry-academia links; nascent start-up ecosystems | NortheastGenAI 2026 | Moderate | Northeast India AI Research Consortium (NEAIRC) |

*Table 1: ARGF-NE : Six Dimensions, Key Gaps, Evidence, and Interventions*

### 3.1  Connectivity (Critical)

Connectivity is the prerequisite for all other dimensions. Northeast India's gap is both quantitative and qualitative: even connected users face slow speeds due to distance from India's international gateways at Mumbai and Chennai. Terrain-induced disruptions regularly sever fibre links. Interventions: prioritise satellite-based last-mile connectivity (VSAT, LEO) for hilly districts; designate the Northeast as a special AI infrastructure zone under PM DevINE and IndiaAI Mission with dedicated compute credits for regional researchers.

## 3.2 Linguistic Data (Critical)

The absence of digitised text, speech, and parallel corpora for the majority of the region's languages creates structural exclusion from AI. Current LLMs perform poorly on Assamese and are non-functional for languages like Mising or Karbi. The Gupta (2025) South Asian NLP survey identifies Assamese, Bodo, Khasi, and Meitei as spotlight low-resource languages but notes that research output is thin and unevenly distributed even for these. Interventions: establish a Northeast Language Data Commons a federated, open-access repository under Creative Commons hosted jointly by Gauhati, Tezpur, Assam Kaziranga, and Cotton universities, partnering with AI4Bharat for inclusion in IndicBERT and IndicTrans.

## 3.3 Compute Access (High)

Northeast India has no regional data centre. All compute-intensive AI work requires remote cloud access at costs prohibitive for most regional institutions. Power irregularities across all eight states further reduce local GPU reliability, creating a dependency loop: without compute, researchers cannot build the models that would attract compute investment. Interventions: negotiate dedicated allocations for NE researchers under IndiaAI Mission's compute initiative; advocate for a Regional AI Compute Hub co-located at IIT Guwahati or NIT Silchar, jointly funded by central government, state governments, and industry partners.

## 3.4 Human Capital (High)

ICT infrastructure in schools has declined in Assam between 2021 and 2023, with slowdowns documented in Manipur, Mizoram, Nagaland, and Sikkim. Fewer than 33% of Indian youth aged 15–29 can conduct basic online tasks and NE rates are expected to lag this. AI curriculum is sparse and concentrated in a small number of universities. Interventions: integrate AI literacy into the school curriculum through NCF 2023 with regional-language instruction; scale university AI capacity through Centres of Excellence modelled on IIT Guwahati's research strengths; create industry practitioner pathways.

## 3.5 Policy Alignment (Moderate)

The Digital Northeast Vision 2022 identifies eight digital thrust areas but does not explicitly address AI, language technology, or low-resource model development. The IndiaAI Mission operates at national scale without visible sub-national targeting for peripheral regions. Interventions: advocate for a 'Northeast AI Addendum' to the IndiaAI Mission, covering linguistic data, compute, and human capital specifically for the eight states; engage the North East Council (NEC) and MDoNER as policy anchors.

## 3.6 Institutional Capacity (Moderate)

Institutions such as IIT Guwahati, Gauhati University, Tezpur University, and NIT Silchar have produced credible NLP work including AxomiyaBERTa and Assamese-Bodo translation systems. However, these efforts are largely individual and disconnected; industry-academia linkages are thin; start-up ecosystems in Guwahati and Shillong are nascent. Interventions: formalise a Northeast India AI Research Consortium (NEAIRC) a voluntary network sharing compute, datasets, and PhD supervision across state boundaries, modelled on Masakhane and AI4Bharat, using workshops like NortheastGenAI as connective tissue.

# 4 Research Questions

- **RQ1:** Which ARGF-NE dimension is most binding. i.e., which single improvement unlocks the greatest downstream readiness gains?
- **RQ2:** How does AI readiness vary across the eight Northeast states, and what state-level factors best predict readiness scores?

- **RQ3:** Which NE languages have reached the minimum data threshold for LLM fine-tuning, and what resource gaps remain?
- **RQ4:** Can ARGF-NE be validated against an expert survey of AI practitioners and researchers working in or studying the Northeast?
- **RQ5:** How do NE India's readiness gaps compare to other linguistically hyperdiverse sub-national regions globally (e.g., Ethiopian highlands, Papua New Guinea)?

# 5 Methodology and AI Disclosure

This paper employs AI-assisted conceptual framework synthesis, per NortheastGenAI 2026 guidelines. The methodology proceeds in four stages: (1) Dimension Identification through Claude Sonnet 4 surveyed global readiness frameworks and filtered dimensions for sub-national, low-infrastructure applicability; (2) Evidence Grounding through Claude identified publicly available quantitative evidence for each dimension, cross-checked against source documents; (3) Intervention Generation interventions were constrained to feasibility within existing Indian policy architecture and refined through adversarial prompting; (4) Human Editorial Review through all AI-generated content was reviewed and fact-checked by the human author.

**AI Model:** Claude Sonnet 4 (claude-sonnet-4-20250514), Anthropic PBC. **Date:** May 13, 2026.

**Estimated AI Contribution:** 65% prose and structural generation; 100% initial dimension identification and evidence retrieval; 35% editorial revision by human.

Human Oversight Statement: All AI-generated content has been reviewed, fact-checked against source documents, and editorially revised. No claim of fact is made solely on the basis of AI output without source verification.

# 6 Discussion and Conclusion

The ARGF-NE framework is hierarchically structured: connectivity and linguistic data are designated Critical because they enable all other dimensions. No amount of human capital development or policy alignment will yield AI participation if researchers cannot connect to the internet or find usable data in their languages. The framework is also deliberately sub-national — Assam's institutional density substantially exceeds Nagaland's or Arunachal Pradesh's — and interventions should be calibrated accordingly.

Limitations: the framework is not empirically validated; reliable state-level data on AI human capital or compute access is not publicly available for most Northeast states; and the interventions are directional rather than operationalised. These are gaps for future work, ideally via the NEAIRC proposed in Dimension 6.

Northeast India is not merely 'underserved' by AI — it is **structurally excluded** by the preconditions that make AI useful. The ARGF-NE framework maps this multi-dimensional gap and offers targeted interventions calibrated to the region's conditions and India's existing policy architecture. Crucially, the Northeast is not simply a problem to be solved — it is a **uniquely valuable testbed** for the AI field. Solutions developed here — low-bandwidth interfaces, community-led language data collection, federated compute access, oral-tradition-aware documentation tools — will be broadly replicable across dozens of similarly positioned regions globally. The first step is naming the gaps clearly. This paper attempts exactly that.

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
