# OpenReview forum: "Beyond Connectivity: A Multi-Dimensional AI Readiness Gap Analysis for Northeast India"
_NortheastGenAI/2026/Workshop — NortheastGenAI 2026 Workshop Submission_

### Official Review · ~Badal_Nyalang1 · 2026-05-23
**Solid framework paper — Accept**

**Rating:** 7
**Confidence:** 4

**Review:**

**Relevance: Strong**
Clean T3 fit. The sub-national AI readiness angle is well-matched to the workshop's purpose and the NE India grounding is consistent and specific throughout.

**Plausibility: Moderate**
The framework is coherent and the evidence citations are real. However several specific numbers are hard to verify — the "<2% of national broadband" figure and the BharatNet gram panchayat counts are plausible but sourced loosely. The paper is honest that no primary data was collected and that interventions are directional rather than operationalised. That honesty is appropriate.

One notable issue: the paper cites "NortheastGenAI 2026" as evidence for institutional capacity gaps. That is circular — the workshop itself cannot be a cited evidence source in a paper submitted to that same workshop.

**Novelty: Moderate**
The six-dimension ARGF-NE framework is the contribution. It is not methodologically novel but is genuinely useful as a structured synthesis. The comparative framing against Oxford and World Bank indices is a reasonable hook.

**Clarity: Strong**
Well structured. Table 1 is the clearest single-table summary of NE India's AI readiness gaps seen in any submission so far. The AI disclosure is the most detailed and technically precise in the batch — model name, version string, date, and contribution percentages all included.

**Verdict: Accept**
Solid exploratory framework paper. The circular citation should be removed before proceedings. Otherwise ready as submitted.

*This review was generated with AI assistance and checked by the workshop chairs.*

---

### Decision · Program_Chairs · 2026-05-23

**Decision:**

Accept

**Comment:**

An exploratory framework paper with a consistent and specific NE India focus. The six-dimension ARGF-NE framework is a useful structured contribution and Table 1 is one of the clearest summaries of AI readiness gaps across the region seen in any submission. The AI disclosure is the most detailed and technically precise in the batch. One issue: the paper cites NortheastGenAI 2026 as an evidence source, which is circular and should be removed before presentation.

Decision: Accept